# Revealing the Impact of Genomic Alterations on Cancer Cell Signaling with an Interpretable Deep Learning Model

**DOI:** 10.3390/cancers15153857

**Published:** 2023-07-29

**Authors:** Jonathan D. Young, Shuangxia Ren, Lujia Chen, Xinghua Lu

**Affiliations:** 1Intelligent Systems Program, School of Computing and Information, University of Pittsburgh, Pittsburgh, PA 15260, USA; shr81@pitt.edu; 2Department of Biomedical Informatics, School of Medicine, University of Pittsburgh, Pittsburgh, PA 15260, USA; luc17@pitt.edu

**Keywords:** somatic genomic alterations, gene expression, pathway, deep learning

## Abstract

**Simple Summary:**

Cancer results from aberrant cellular signaling caused by somatic genomic alterations (SGAs). However, inferring how SGAs cause aberrations in cellular signaling and lead to cancer remains challenging. We designed an interpretable deep learning model to encode the impact of SGAs on cellular signaling systems (represented by hidden nodes in the model) and eventually on tumor gene expression. The transparent deep learning architecture enabled the model to discover drivers affecting common signaling pathways and partially resolve the causal structure of signaling proteins. This is an early attempt to use transparent deep learning model, in contrast to conventional "black box" approach, to learn interpretable insights into cancer cell signaling systems. A better representation of signaling system of a cancer cell sheds light on the disease mechanisms of the cancer and can guide precision medicine.

**Abstract:**

Cancer is a disease of aberrant cellular signaling resulting from somatic genomic alterations (SGAs). Heterogeneous SGA events in tumors lead to tumor-specific signaling system aberrations. We interpret the cancer signaling system as a causal graphical model, where SGAs affect signaling proteins, propagate their effects through signal transduction, and ultimately change gene expression. To represent such a system, we developed a deep learning model called redundant-input neural network (RINN) with a transparent redundant-input architecture. Our findings demonstrate that by utilizing SGAs as inputs, the RINN can encode their impact on the signaling system and predict gene expression accurately when measured as the area under ROC curves. Moreover, the RINN can discover the shared functional impact (similar embeddings) of SGAs that perturb a common signaling pathway (e.g., PI3K, Nrf2, and TGF). Furthermore, the RINN exhibits the ability to discover known relationships in cellular signaling systems.

## 1. Introduction

The cellular mechanisms leading to cancer in an individual are heterogeneous, nuanced, and not well understood. It is well appreciated that cancer is a disease of aberrant signaling, and the state of a cancer cell can be described in terms of abnormally functioning cellular signaling pathways. Precision oncology depends on the ability to identify the abnormal cellular signaling pathways causing a patient’s cancer, so that patient-specific effective treatments can be prescribed—including targeting multiple abnormal pathways during a treatment regime. Aberrant signaling in cancer cells usually results from somatic genomic alterations (SGAs) that perturb the function of signaling proteins. Although large-scale cancer genomic data are available, such as those from The Cancer Genome Atlas (TCGA) and the International Cancer Genome Consortium (ICGC), it remains a very difficult and unsolved task to reliably infer how the SGAs in a cancer cell cause aberrations in cellular signaling pathways based on the genomic data of a tumor. One challenge is that the majority of the genomic alterations observed in a tumor are non-consequential (*passenger* genomic alterations) with respect to cancer development and only a few are *driver* genomic alterations, i.e., genomic alterations that cause cancer. Furthermore, even if the driver genomic alterations of a tumor are known, it remains challenging to infer how aberrant signals of perturbed proteins affect the cellular system of cancer cells, because the states of signaling proteins (or pathways) in the signaling system are not measured (latent). This requires one to study the causal relationships among latent (i.e., hidden or unobserved) variables, which represent the state of individual signaling proteins, protein complexes, or certain biological processes within a cell, in addition to the observed variables in order to understand the disease mechanisms of an individual tumor and identify drug targets. Most causal discovery algorithms have been developed to find the causal structure and the parameterization of the causal structure relative to the *observed* variables of a dataset [1,2,3,4,5,6]. Only a small number of causal discovery algorithms also find the latent causal structure [7,8,9,10,11]. Interestingly, Xie et al. [11] and Huang et al. [10] developed algorithms to learn hierarchical structures among latent variables based on compositional statistical structures. However, due to the large search space, these algorithms are not suitable for handling high-dimension data, which we deal with in our study.

Deep learning represents a group of machine learning strategies, based on neural networks, that learn a function mapping inputs to outputs. The signals of input variables are processed and transformed with many hidden layers of latent variables (i.e., hidden nodes) [12,13,14]. These hidden layers learn hierarchical or compositional statistical structures, meaning that different hidden layers capture structures of different degrees of complexity [9,15,16,17]. Researchers have previously shown that deep learning models can represent the hierarchical organization of signaling molecules in a cell [18,19,20,21,22], with latent variables as natural representations of unobserved activation states of signaling molecules (e.g., membrane receptors or transcription factors). However, deep learning models have not been broadly used as tools to infer *causal relationships* in a computational biology setting, partly due to deep learning’s “black box” nature.

Relevant to the work presented in this paper, here, we briefly describe some of the studies that used neural-network-based approaches to discover gene regulatory networks (GRNs). A study published in 1999 by Weaver and Stormo [23] modeled the relationships in gene regulatory networks as coefficients in weight matrices (using the familiar concepts of weighted sums and an activation function in their model). However, they only tested their algorithm on simulated time-series data, as large gene expression datasets were not yet available and performing these calculations on large numbers of data was quite challenging at the time. Similarly, Vohradsky [24] and Keedwell et al. [25] also interpreted the weights of a quasi-recurrent neural network as relationships in gene regulatory networks. Again, simulated time-series data were used in these studies with reasonable results. A more recent study published in 2015 [26] used a linear classifier with one input “layer” and one output “layer” (which they called a neural network) to infer regulatory relationships (as represented by the weights of the weight matrix in the linear classifier) among genes in lung adenocarcinoma gene expression data. However, they did not evaluate their learned regulatory network, did not use DNA mutation data (as we do in this work), and did not use neural networks. There have also been studies that used genetic algorithms (evolving a weight matrix of regulatory pathways) to infer gene regulatory networks [27]. Like the studies above, this work was ahead of its time, and they were only able to test their algorithm on simulated expression data. Improving upon Ando and Iba [27], Keedwell and Narayanan [28] used the weights representing a single-layer artificial neural network (ANN) (trained with gradient descent) to represent regulatory relationships among genes in gene expression data. Interestingly, they also used a genetic algorithm for a type of feature selection to make the task more tractable. They achieved good results on simulated temporal data and also tested their method on real temporal expression data (112 genes over nine time points) but did not have ground truth for comparison. In another study, Narayanan et al. [29] used single-layer ANNs to evaluate real, non-temporal gene expression data in a classification setting (i.e., they did not recover GRNs). However, the high dimensionality of the problem was again a major limiting factor.

Many of the studies discussed above used the weights of a neural network to represent relationships in gene regulatory networks. However, it does not appear that these studies attempted to use a neural network to identify latent causal structures at different hierarchical levels (i.e., cellular signaling system), as we do in this work. In general, the above studies used very basic versions of neural network (without regularization limiting the magnitude of the weights and thereby the complexity of the learned function) and, due to computing constraints and the absence of large genomic datasets, were unable to train their networks on high-dimension data. Also, most of the methods above require temporal data, as opposed to the static genomic data that we utilize here. Overall, none of the studies discussed above used a deep neural network (DNN) to predict expression data from genomic alteration data and then recover causal relationships in the weights of a DNN, as we do in this paper.

More recent work from our group used deep learning to simulate cellular signaling systems that were shared by human and rat cells [18] and to recover components of the yeast cellular signaling system, including transcription factors [19]. These studies utilized unsupervised learning methods, in contrast to the supervised methods used in this study. Also, the previous studies by our group did not attempt to find causal relationships representing the cellular signaling system.

In a recent work [9], we developed a deep learning algorithm, named redundant-input neural network (RINN), to learn causal relationships among latent variables from data inspired by cellular signaling pathways. The RINN solves the problem where a set of input variables *cause* the change in another set of output variables, and this causal interaction is mediated by a set of an unknown number of latent variables. The constraint of inputs causing outputs is necessary to interpret the latent structure as causal relationships (see [9] for more details regarding the causal assumptions of the RINN). A key innovation of the RINN model is that it is a partially transparent model, allowing input variables to directly interact with all latent variables in its hierarchy. This allows the RINN to constrain (in conjunction with $L_{1}$ regularization) an input variable to be connected to a set of latent variables that can sufficiently encode the impact of the input variable on the output variables. In Young et al. [9], we showed that the RINN outperformed other algorithms, including neural-network-based algorithms and a causal discovery algorithm known as DM (Detect MIMIC (Multiple Indicators, Multiple Input Causes)), in identifying latent causal structures in various types of simulated data.

In the current study, we took advantage of the partially transparent nature of the RINN model and used the model to learn a representation of the cancer cellular signaling system. In this setting, we interpreted the cellular signaling system as a hierarchical causal model of interactions among the activation states of proteins or protein complexes within a cell. Based on the assumption that the somatic genome alterations (SGAs) that drive the development of a cancer often influence gene expression, we trained the RINN on a large number of tumors from TCGA using tumor SGAs as inputs to predict cancer differentially expressed genes (DEGs) (outputs). We then evaluated the latent structure learned with the hidden layers of the RINN in an attempt to learn components of the cancer cellular signaling system. We show that the model is capable of detecting the shared functional impact of SGAs affecting members of a common pathway. We also show that the RINN can capture cancer signaling pathway relationships within its hidden variables.

## 2. Results

### 2.1. Overview of the RINN Model

An RINN shares similarities with a regular supervised feed-forward deep neural network but incorporates a modified architecture. Unlike a conventional network, an RINN not only establishes a fully connected link between the input and the first hidden layer but also includes connections from the input to every subsequent hidden layer (Figure 1). This unique design enables the RINN to acquire knowledge about direct causal connections between an input SGA and any hidden node within the network.

### 2.2. Model Selection

We trained approximately 23,000 RINN and 23,000 DNN models on the TCGA training dataset with distinct sets of hyperparameters (e.g., number of hidden nodes, activation function, regularization rate, etc.). We evaluated each trained model on multiple validation datasets to evaluate how well it performed. We hypothesized that the models with the most parsimonious weight structure, while still maintaining the ability to accurately capture the statistical relationship between SGAs and DEGs, likely learned optimal representations of the impact of SGAs on cancer cells. To reflect the balance between these two objectives, we visualized the performance of all models on a scatter plot with the validation set error and the sum of the absolute value of all weights as axes (Figure 2). Each blue dot in this figure represents a neural network trained on a unique set of hyperparameters. We ranked models based on their Euclidean distance from the origin ($d_{x}$), and models with the shortest $d_{x}$ were selected as the models with the best balance between sparsity and validation set error.

The ten best RINN models (i.e., models with the lowest $d_{x}$) are shown in Table 1. Among the multiple activation functions studied, the softplus activation function overwhelmingly provided the best results. This was also seen with the best DNN models. Overall, the number of hidden nodes in each hidden layer for the RINNs were relatively small (∼100) compared with the dimensionality of output space (5259). Despite being initialized with eight hidden layers, the best RINN models utilized only three or four of the eight hidden layers, indicating that the model could automatically trim hidden layers (assigning 0 values to hidden nodes) due to regularization. All top ten DNN models had two hidden layers, with sizes ranging from 50 to 644 hidden nodes.

### 2.3. Predicting DEG Status with Given SGAs

We examined how well deep learning models could predict DEGs with SGAs given as inputs using different metrics, including cross-entropy loss, AUROC, and AUPR. Cross-validation model selection metrics for RINNs and DNNs are compared in Table 2. Table 2 shows the mean and standard deviation of the metrics on all 5259 DEGs. In general, RINNs and DNNs performed similarly across all metrics. RINNs and DNNs performed significantly better than *k*-nearest neighbors or random controls. The best RINN and DNN models according to the shortest $d_{x}$ (meaning the most regularized) performed similarly but slightly worse than the much less regularized RINNs and DNNs. $\sum_{1}^{8}$$|W_{i}|$ is a surrogate measure of the density of a neural network, with higher values indicating higher density of edges, in general.

To obtain a better idea of how well the models performed in predicting individual DEGs from SGAs, we plotted DEG AUROC histograms for individual models and relevant control models (Figure 3). Figure 3 shows the AUROC values for all 5259 DEGs in the best RINNs and DNNs. RINNs and DNNs vastly outperformed the control models with much higher AUROC values. Interestingly, for many DEGs, the models could achieve AUROC values greater than 0.8. Again, DNNs and RINNs performed similarly when compared to each other. Importantly, there was not a large difference in the AUROC curves among RINN (and DNN) models selected according to the lowest CEL (cross-entropy loss) and those selected using the shortest ED (Euclidean distance), despite the models selected according to shortest ED being much more regularized.

### 2.4. Learning Shared Functional Impacts of SGAs Using RINNs

An RINN allows an SGA to interact with all hidden nodes in the latent hierarchy. After training with regularization, the majority of the weights from SGAs to hidden nodes were set close to 0.0; thus, the remaining weights reflected how the impact of the SGAs was propagated through the latent variables and eventually influenced gene expression. In other words, these remaining weights (edges) between the SGAs and hidden nodes served as signatures (or vector embeddings) representing an SGA’s functional impact. We set out to assess whether the weight signatures learned by the RINNs truly reflected the functional impact of SGAs. Here, one can measure the closeness of the functional impacts of two SGAs by using cosine-angle similarity. We hypothesized that if the RINN correctly captures the functional impact of SGAs, then the weight signature of a gene (SGA) should be more similar to genes belonging to a common pathway than to genes from other pathways. To this end, we used the well-documented cancer pathways recently reported by the TCGA pan-cancer analysis project [30] (Appendix A) to determine which genes shared a common pathway.

For each of the 35 driver SGAs that are in the known pathways from the TCGA pan-cancer study [30] (and that are also in our SGA dataset), we calculated the cosine similarity of its weight signature with respect to all other SGA weight signatures and listed the three most similar SGAs in Table 3. Clearly, many of the SGAs that had high cosine similarity relative to the query SGA were in the same pathway as the query SGA according to [30]. Most of the SGAs in the PI3K pathway had high cosine similarity to the other SGAs in the PI3K pathway. *KEAP1* and *NFE2L2* (the only members of the Nrf2 pathway in our SGA dataset) were both found to be each other’s SGA with the highest cosine similarity (i.e., each other’s closest neighbor). Also, *SMAD4* and *ACVR2A* (only members of the TGFβ pathway in the SGA dataset) were also found to be each other’s SGA with the highest cosine similarity. Additional cosine-similarity relationships were found between members of the same pathway for the remaining pathways, but these relationships were less frequent than the results for PI3K, Nrf2, and TGFβ mentioned above. These cosine-similarity results suggest that the hidden nodes in our trained RINNs may represent biological entities or components of a cellular signaling system involved in propagating signals of SGA events, indicating that the connectivity found in our trained RINNs is related to cellular signaling pathways.

To compare the RINNs and DNNs, we also performed the same cosine-similarity experiment of Table 3 but with weights from the best three DNN models (Table 4). For this experiment, we only used weights from the input SGAs to the first hidden layer, as these are the only weights in a DNN that are specific to individual SGAs. In Table 4, many of the SGAs that had high cosine similarity relative to the query SGA were in the same pathway as the query SGA according to [30]. Overall, there are 28 genes in bold in Table 3 (RINN) and 27 genes in bold in Table 4 (DNN). The DNNs performed marginally worse than the RINNs in capturing the relationships in the pathways, mostly because the DNNs failed to robustly capture as much of the shared function of the SGAs affecting the PI3K pathway as the RINNs. However, the DNNs performed slightly better in capturing pathway relationships in the RTK/RAS pathway (especially with cosine similarity relative to *FGFR3*). Just as with the RINNs, *KEAP1* and *NFE2L2* were both found to be each other’s SGA with the highest cosine similarity with the DNNs. *SMAD4* and *ACVR2A* were also found to be similar to one another but not as highly similar as with the RINNs. These results indicate that when a DNN can accurately predict DEGs (outputs) from SGAs (inputs), the DNN weight-signature cosine similarity can also capture the functional similarity of SGAs.

An alternative approach to illustrating the information provided by SGA weight signatures is to visualize the highest cosine similarity between weight signatures as edges in a graph and then apply a community discovery algorithm to identify closely related SGAs (Figure 4). In Figure 4, the discovered communities are shown as nodes and edges of the same color. In this figure, a directed edge connects an SGA (at the tail) to another SGA (at the head) that has high cosine similarity relative to the SGA at the tail of the edge; a bi-directed edge indicates that two SGAs are mutually highly similar; the thickness of an edge represents the amount of cosine similarity; and the size of the SGA node represents the degree of that node. Figure 4A is a visualization of the highest cosine similarity for each SGA, and Figure 4B is a visualization of the three highest cosine-similarity results for each SGA. The nodes and edges are colored according to the community after running a community detection algorithm. Many of the communities that were discovered with this procedure correspond to the pathways in Appendix A. For example, the green community in Figure 4A captures the majority of the PI3K pathway; the dark-blue community and brown community capture the TGFβ and Nrf2 pathways, respectively; and the purple community captures many of the genes (SGAs) in the Wnt, Hippo, and Notch pathways. In Figure 4B, the majority of the PI3K pathway is part of the green community and a blue community emerges with many members of the cell-cycle pathway. Many other additional pathway relationships within the ground-truth pathways are seen in Figure 4, reinforcing the hypothesis that the latent structure of trained RINNs contains cellular signaling pathway relationships.

We also visualized the weights from SGAs to hidden nodes (i.e., weight signatures) as a heatmap, where each column represents an SGA, each row represents a hidden node, and the weight from an SGA to a hidden node is represented as an element in the color-coded heatmap. We performed hierarchical clustering on the weight signatures derived from a single RINN model (the RINN with the shortest $d_{x}$) as shown in Figure 5. Within this clustering and the dendrogram, we observe many relationships that are present in the ground-truth pathways. For example, the green cluster contains much of the PI3K pathway; the orange cluster captures the TGFβ pathway; the purple cluster contains the two members of the Nrf2 pathway; and the gray cluster contains the members of the Notch pathway and two of the three members of the Hippo pathway.

In addition to the above clustering relationships, the heatmap also illustrates the ability of the RINN to automatically determine the “optimal” number of hidden layers that were needed to encode information from SGAs to DEGs. Here, the hidden layers are numbered starting from the inputs (SGAs) to the outputs (DEGs). This heatmap shows that all the weights before hidden layer 5 had a value of zero, and only one hidden node in layer 5 had incoming weights with nonzero values. All of the top ten models (including this one) utilized only three or four hidden layers, similar to what is seen in Figure 5. Hidden layer 7 seemed to have the most nonzero incoming weights. Interestingly, many of the most important cancer-related SGAs (e.g., *EGFR*, *TP53*, *CDKN2A*, *APC*, *PIK3CA*) had a larger number of weights and more largely valued weights than other SGAs in Figure 5, suggesting that this model captured relevant biological information.

### 2.5. Inferring Causal Relationships Using RINNs

One of the goals of this work was to discover latent causal structures relevant to cancer cellular signaling pathways. We hypothesized that RINNs can reveal the causal relationships among the proteins perturbed by SGAs. For example, if SGA (intervention) $i_{1}$ is connected to latent variable $h_{1}$ and another SGA, $i_{2}$, is connected to another latent variable, $h_{2}$, and if $h_{1}$ is connected directly upstream of $h_{2}$ with a nonzero-weighted edge in the latent hierarchy, it would suggest that the signal of $i_{1}$ is upstream of that of $i_{2}$ and that the protein affected by $i_{1}$ causally regulates the protein affected by $i_{2}$ in the cellular system. To examine this type of relationship, we visualized the relationships among the SGAs from four TCGA pathways via the latent nodes they were connected to, as a means to search for causal structures revealed by the RINN models (Figure 6). Overall, the learned causal graphs were more complex and denser than the graphs from the pan-cancer TCGA analysis (Appendix A), in that an SGA in RINNs was often connected to multiple latent variables (depending on the cutoff threshold). This suggests that there is still some difficulty in directly interpreting the weights of an RINN as causal relationships with the current version of the algorithm. It may also suggest that the ground truth we used here is not detailed enough for our purposes, or may have some inaccuracies, and thus is more of a “silver standard”. In general, the learned causal graphs had a large number of false-positive edges (when compared with Appendix A) and multiple instances of redundant causal edges—where a single SGA acted multiple times on the same path.

However, some true causal relationships could be seen within these graphs. For example, *both* Models 0 and 1 had a directed path as follows: $i_{KEAP1}\to h_{1}\to h_{2}\leftarrow i_{NFE2L2}$, where *i* represents an intervention (i.e., SGA). If we assume that $h_{1}$ is a representation of the *KEAP1* protein and $h_{2}$ is a representation of the *NFE2L2* protein, then we can interpret the above path as *KEAP1*→*NFE2L2*, i.e., *KEAP1* causes or changes the state of *NFE2L2*. This is consistent with the Nrf2 pathway shown in Appendix A. Model 2 had the following directed path: *KEAP1*$\to h_{1}\leftarrow$*NFE2L2*; this has a more ambiguous interpretation but does emphasize the correlation between these two SGAs. Also, it is plausible that if the weight threshold were decreased slightly for Model 2, there would be an edge between the green hidden node and the blue hidden node, meaning that all three models would then capture the *KEAP1*→ *NFE2L2* causal relationship. This emphasizes the importance of finding a more robust means of thresholding in future work.

All the RINNs represented in Figure 6 (and Figure 5) only utilized three or four hidden layers to learn the function mapping SGAs to DEGs. Because we used $L_{1}$ regularization, RINNs learned to only use the necessary number of hidden layers to perform well in the prediction task. For example, if only three hidden layers were needed to perform well in prediction, the regularized objective function would set all the weights before the last three hidden layers to 0.0. We set the number of hidden layers to eight, as we hypothesized that eight hidden layers was enough hierarchical complexity to capture cancer cellular signaling pathways in a meaningful way. Figure 5 and Figure 6 suggest that this was a correct assumption in this setting for the data we used. As with other model selection techniques, if our trained RINNs had utilized all eight layers during model selection, we could have easily increased the number of hidden layers to more than eight and perform another round of model selection.

### 2.6. Shared Hidden Nodes across Top Ten Models

As shown in Figure 6, SGAs perturbing members of a common pathway were closely connected to a similar set of hidden variables, and often many SGAs were directly connected to a common hidden node. This suggests that such hidden nodes encoded a common signal that was shared by the SGAs. We hypothesized that if SGAs from a pathway were connected to a similar set of shared hidden nodes in multiple, different RINN models, this would have indicated that the RINNs could repeatedly detect the common impact of the SGAs and thus use a common set of hidden nodes to encode their impact with respect to DEGs. In other words, an RINN consistently encodes the shared functional impact of SGAs perturbing a common pathway, and the function of these hidden nodes in a specific RINN model becomes partially transparent. We labeled hidden nodes based on their SGA ancestors (see the Visualizing an RINN as a Causal Graph section) and then examined whether such hidden nodes were conserved across models (Table 5). Indeed, many hidden nodes were conserved across the top ten models. More specifically, Table 5 shows the number of models that shared the labeling of a specific hidden-node and whether or not this would be expected by chance. For example, a hidden-node mapping to all five members of the PI3K pathway, {AKT1,PIK3CA,PIK3R1,PTEN,STK11}, was found in all of our top ten models. Given five random SGAs, the expected number of the top 10 models to share a hidden node mapped to all five random SGAs was 0.0 models. This means that in our 30 replicates of random controls with five random SGAs, none of the top ten models ever shared a hidden node mapped to all five random SGAs. This also means that the above result for all five members of the PI3K pathway is a very strong result and a definite pathway relationship was discovered with the RINN setup in this work. Many of the four-SGA and three-SGA labeled hidden nodes were also found in many more models than the corresponding random control, such as the {AKT1,PIK3CA,PTEN,STK11}, {PIK3CA,PIK3R1,PTEN,STK11}, {AKT1,PIK3CA,STK11}, {PIK3R1,PTEN,STK11}, and {PIK3CA,PIK3R1,PTEN} labeled hidden nodes that were found.

A hidden node mapped to both members of the Nrf2 and TGFβ pathways was also found in all ten models, which was much higher than the number of models expected given two random SGAs. Given random SGAs, only 0.7 ± 1.2 models of the top 10 were expected to share the same two-SGA labeled hidden node. Also, a hidden node mapped to all three members of the Notch pathway was shared by nine of the top ten models, which was also well beyond the number of models expected given random SGAs.

## 3. Discussion

In this study, we show that deep learning models, RINNs and DNNs, can capture the statistical relationships between genomic alterations and transcriptomic events in tumor cells with reasonably high accuracy, despite the small number of training cases relative to the high dimensionality of the data. Our findings further indicate that a regularized deep learning model with redundant inputs (i.e., RINN) can capture cancer signaling pathway relationships within its hidden variables and weights. The RINN models correctly captured much of the functional similarity among SGAs that perturb a common signaling pathway, as reflected by the SGAs’ similar interactions with the hidden nodes of the RINN models (i.e., cosine similarity of SGA weight signatures). This shows that SGAs in the same pathway share similar interactions (in terms of connection and weights) with a set of latent variables. These are very encouraging results for eventually using a future version of the RINN to find signaling pathways robustly. Many of the most well-known cancer driver genes (*EGFR*, *TP53*, *CDKN2A*, *APC*, and *PIK3CA*) were found to have dense SGA weight signatures and weights with larger values relative to the other genes we analyzed, reinforcing the importance of these genes in driving cancer gene expression and the validity of our models. Our results indicate that an RINN consistently employs certain hidden nodes to represent the shared functional impact of SGAs perturbing a common pathway, although different instantiations of the RINN could use totally different hidden nodes. The ability of an RINN to explicitly connect SGAs and hidden nodes throughout the latent hierarchy essentially makes the RINN a partially transparent deep learning model, so that one can interpret which hidden nodes encode and transmit the signals (i.e., functional impact) of SGAs in cancer cells. Finally, we show that RINNs are capable of capturing some causal relationships (given our interpretation of the hidden nodes) among the signaling proteins perturbed by SGAs. All these results indicate that by allowing SGAs to directly interact with hidden nodes in a deep learning model, the RINN model provides constraints, information, and flexibility to enable certain hidden nodes to encode the impact of specific SGAs.

Overall, both RINNs and DNNs are capable of capturing statistical relationships between SGAs and DEGs. However, latent variables in a DNN model (except those directly connected to SGAs) are less interpretable because the latent variable information deeper in the network is more convoluted. In a DNN model, all SGAs have to interact with the first layer of hidden variables, and their information is then propagated through the whole hierarchy of the model. In such a model, it is difficult to pinpoint how the signal of each SGA is propagated. Hidden layers in a DNN are alternate representations of all the information in the input necessary to calculate the output, whereas each hidden layer of an RINN does not need to capture all the information in the input necessary to predict the output because there are multiple chances to learn what is needed from the input (i.e., redundant inputs). This difference gives RINNs more freedom in how to choose to use the information in the input. The redundant inputs of an RINN represent an attempt to deconvolute the signal of each SGA by giving the model more freedom to take advantage of the hierarchical structure and choose latent variables at the right level of granularity to encode the signal of an SGA, e.g., early in the network in the first hidden layer or in later layers in the network. This approach is biologically sensible because different SGAs do affect proteins at different levels in the hierarchy of the cellular signaling system. It is expected that an SGA perturbing a transcription factor (e.g., *STAT3*) impacts a relatively small number of genes in comparison to an SGA that perturbs at a high level in the signaling system (e.g., *EGFR*). Refined granularity enables RINNs to search for the "optimal" structure in order to encode the signaling between SGAs and DEGs while satisfying our sparsity constraints, leading to RINNs with three to four relatively sparsely connected layers of hidden variables; whereas DNNs tend to use two layers of relatively densely connected latent variables.

A DNN cannot capture the same causal relationships that an RINN can. By nature of its architecture and design, a DNN can only capture *direct* causal relationships (i.e., edges starting from an observed variable) between the input and the first hidden layer—DNNs cannot capture direct causal relationships between the input and any other hidden layers. This means that a DNN cannot be used to generate causal graphs like the ones shown in Figure 6. In addition, a DNN cannot capture causal relationships among *input SGAs* as described in the Inferring Causal Relationships Using RINNs section, meaning that one cannot infer the causal relationships among SGAs with a DNN. This is because DNNs do not have edges between hidden nodes in the same layer (or redundant inputs). Let us consider $path_{KEAP1}$ and $path_{NFE2L2}$ as the paths, with *KEAP1* and *NFE2L2* as the source nodes, respectively, in a DNN. In a DNN, to determine that there is a dependency between *KEAP1* and *NFE2L2*, eventually, these two paths would need to collide on a hidden node. When these two paths collide on a hidden node, the number of edges in each path will be the same, meaning that the direction of the causal relationship is ambiguous. This limitation of DNNs can be remedied by adding redundant inputs (i.e., RINN). Using the RINN architecture allows us to infer order to the causal relationships among SGAs; this design difference and the extension of causal interpretability are what sets RINNs apart from DNNs.

It is intriguing to further examine whether the hierarchy of hidden nodes can capture causal relationships among the signals encoded by SGA-affected proteins. We have shown that many of the pathway relationships and some known causal relationships were present in the hierarchy reflected by the weight matrices of our trained models. However, we also noticed that in our RINN models, an SGA was often connected to a large number of hidden nodes, which were in turn connected to a large number of other hidden nodes—meaning that the current RINN model learns relatively dense causal graphs. While one can infer the relationships between the signal perturbed by distinct SGAs of a pathway, our current model cannot directly output a causal network that looks like those commonly shown in the literature. We plan to develop the RINN into an algorithm that is able to find more easily interpretable cellular signaling pathways when trained on SGA and DEG data. The following algorithm modifications will potentially lead to better results in the future: (1) incorporating differential regularization of the weights, (2) using constrained and parallelized versions of evolutionary algorithms to optimize the weights and avoid the need to threshold weights, and (3) training an autoencoder with a bottleneck layer to encourage hidden nodes to more easily represent biological entities and then using these weights (and architecture) to initialize an RINN.

In order to interpret the weights of a neural network as causal relationships among biological entities, we assume that the causal relationships among biological entities can be approximated with a linear function combined with a simple nonlinear function (e.g., activation(wx+b)), where all variables have scalar values and activation represents a simple nonlinear function such as ReLU or softplus. This is a necessary assumption in order to interpret all nonzero hidden nodes as biological entities; however, it could also be the case that some hidden nodes are not biological entities but rather some intermediate calculation required to compute a relationship among biological entities that cannot be modeled with activation(wx+b). Given the high density of the models learned with TCGA data, it is possible that the relationships among some biological entities cannot be modeled with activation(wx+b), suggesting that more complex activation functions are needed or that biological entities may be present in every other hidden layer. It would be interesting to explore using more complex activation functions and specifically using an unregularized one in a hidden-layer neural network as an activation function for each hidden node in an RINN. This setup would account for even quite complex relationships among biological entities captured as latent variables. See [9] for additional discussion of this topic.

A cellular signaling system is a complex information-encoding/-processing machine that processes signals arising from extrinsic environmental changes or perturbations (genetic or pharmacological) affecting the intrinsic state of a cell. The relationships of cellular signals are hierarchical and nonlinear by nature, and deep learning models are particularly suitable for modeling such a system [18,19,20,21,22]. However, conventional deep learning models behave like “black boxes”, such that it is almost impossible to determine what signal a hidden node encodes, with few exceptions in image analysis, where human-interpretable image patterns can be represented with hidden nodes [15,16,17]. Here, we took advantage of our knowledge of cancer biology that SGAs causally influence the transcriptomic programs of cells, and we adopted a new approach that allows SGAs to directly interact with hidden nodes in an RINN. We conjecture that this approach forces hidden nodes to explicitly and thus more effectively encode the impact of SGAs on transcriptomic systems. This hypothesis is supported by the discoveries of this paper that SGAs in a common pathway share similar connection patterns to hidden nodes and that there are hidden nodes that are connected to multiple members of a pathway in different instances of the model. Essentially, our approach also allows certain hidden nodes to be “labeled” and “partially interpretable”. An interpretable deep learning model provides a unique opportunity to study how cellular signals are encoded and perturbed under pathological conditions. Understanding and representing the state of a cellular system further opens directions for translational applications of such information, such as predicting the drug sensitivity of cancer cells based on the states of their signaling systems. To our knowledge, this is the first time that a partially interpretable deep learning model has been developed and applied to study cancer signaling, and we anticipate this approach laying a foundation for developing future explainable deep learning models in this domain.

## 4. Experimental Procedures

### 4.1. Data

The data used in this paper were originally downloaded from TCGA [31,32]. RNA Seq, mutation, and copy number variation (CNV) data over multiple cancer types were used to generate two binary datasets. A binary differentially expressed gene (DEG) dataset was created by comparing the expression value of a gene in a tumor against the distribution of the expression values of the gene across normal samples from the same tissue of origin. A gene was deemed a DEG in a tumor if its value was outside the 2.5% percentile on either side of the normal sample distribution; then, that gene’s value was set to 1. Otherwise, the gene’s value was set to 0. A somatic genome alteration (SGA) dataset was created using mutation and CNV data. A gene was deemed to be perturbed by an SGA event if it hosted a non-synonymous mutation, small insert/deletion, or somatic copy number alteration (deletion or amplification). If perturbed, the value in the tumor for that gene was set to 1, otherwise the value was set to 0.

We applied the tumor-specific causal inference (TCI) algorithm [32,33] to these two matrices to identify the SGAs that causally influence gene expression in tumors and the union of their target DEGs. TCI is an algorithm that finds causal relationships between SGAs and DEGs in each individual tumor, without determining how the signal from the SGA is propagated in the cellular signaling system [32,33]. We identified 372 SGAs that were deemed driver SGAs, as well as 5259 DEGs that were deemed target DEGs of the 372 SGAs using TCI. Overall, combining data from 5097 tumors led to two data matrices (with dimensions of 5097 ×372 and 5097 ×5259) as inputs and outputs for the RINN, where SGAs (inputs) were used to predict DEGs (outputs).

### 4.2. Deep Learning Strategies: RINN and DNN

In this study, we used two deep learning strategies: RINN and DNN. A DNN is a conventional supervised feed-forward deep neural network. A DNN learns a function mapping inputs (***x***) to outputs (***y***) according to
(1)$$\begin{array}{c} f\left(x\right)=\phi \left(...\phi \left(\phi \left(x\cdot W_{1}\right)\cdot W_{2}\right)...\cdot W_{n}\right)=\hat{y} \end{array}$$
where $W_{i}$ represent the weight matrices between layers of a neural network, ϕ is some nonlinear function (i.e., an activation function such as ReLU, softplus, or sigmoid), · represents vector–matrix multiplication, and $\hat{y}$ is the predicted output. This function represents our predicted value for y and, in other words, represents a vector–matrix multiplication followed by the application of a nonlinear function repeated multiple times. An iterative procedure (stochastic gradient descent) is used to slowly change the values of all $W_{i}$ to bring $\hat{y}$ closer and closer to y (hopefully). The left side of Figure 1, without the redundant-input nodes and corresponding redundant-input weights, represents a DNN. DNNs also have bias vectors added to each layer (i.e., to each $h_{i-1}W_{i}$, where $h_{i-1}$ is the previous layer’s output), which are omitted in the above equation for clarity. For a more detailed explanation of DNNs, please see [13].

We introduced the RINN for latent causal structure discovery in [9]. An RINN is similar to a DNN with a modification of the architecture. An RINN not only has the input fully connected to the first hidden layer but also has copies of the input fully connected to each additional hidden layer (Figure 1). This structure allows an RINN to learn direct causal relationships between an input SGA and any hidden node in the RINN. Forward propagation with an RINN is performed as it is with a DNN with multiple vector–matrix multiplications and nonlinear functions, except that an RINN has hidden layers concatenated to copies of the input (Figure 1). Each hidden layer of an RINN with redundant inputs is calculated according to
(2)$$\begin{array}{c} h^{\left(i\right)}=\phi \left(\left(\left[h^{\left(i-1\right)},x\right]\cdot W_{i}\right)+b_{i}\right) \end{array}$$
where $h^{\left(i-1\right)}$ represents the previous layer’s output vector, x is the vector input to the neural network, $\left[h^{\left(i-1\right)},x\right]$ represents concatenation into a single vector, $W_{i}$ represent the weight matrix between hidden and redundant nodes in layer i−1 and hidden layer *i*, ϕ is a nonlinear function (e.g., ReLU), · represents vector–matrix multiplication, and $b_{i}$ represents the bias vector for layer *i*. In contrast to an RINN, a plain DNN calculates each hidden layer as
(3)$$\begin{array}{c} h^{\left(i\right)}=\phi \left(\left(h^{\left(i-1\right)}\cdot W_{i}\right)+b_{i}\right) \end{array}$$

The backpropagation of errors and stochastic gradient descent with RINNs are the same as with DNNs but with additional weights to be optimized. For much more detailed information about the RINN, we recommend that you see our paper where we introduce it [9]. In addition to the architecture modification, all RINNs in this study importantly included $L_{1}$ regularization of the weights as part of the objective function. The other component of the objective function was the binary cross-entropy error. All DNNs also used $L_{1}$ regularization of weights plus binary cross-entropy error as the objective function to be optimized. All RINNs used in this study had eight hidden layers (with varying sizes of the hidden layers), as we hypothesized that cancer cellular signaling pathways do not have more than eight levels of hierarchy. However, this assumption can be easily updated if found to be false.

### 4.3. Model Selection

We hypothesized that since the RINN mimics the signaling processes through which the SGAs of a tumor exert their impact on gene expression (DEGs), the better an RINN model captures the relationships between SGAs and DEGs, the more closely the RINN can represent the signaling processes of cancer cells, i.e., the causal network connecting SGAs and DEGs. We performed an extensive amount of model selection to search for the structures and parameterization that best balanced loss and sparsity or, in other words, the models that fitted the data well. To this end, we performed model selection over 23,000 different sets of hyperparameters (23,000 for RINNs and 23,000 for DNNs).

Model selection was performed using 3 folds of 10-fold cross-validation. Using 3 folds of 10-fold cross-validation gave us multiple validation datasets so we avoided chance overfitting, which can occur with a single validation set. This setup also allowed us to train the models on 90% of the data (and validate them on 10%) for each split of the data, which is important, considering the small number of instances relative to the number of output DEGs that we were trying predict. Importantly, using 3 folds of 10-fold cross-validation takes significantly less time than training on all 10 folds. The running time was especially important in this study because of the large number of hyperparameter sets to be evaluated. All metrics recorded in this study (i.e., AUROC, cross-entropy error, etc.) represented the mean across the three validation sets.

The hyperparameters used in this study included learning rate, regularization rate, number of training epochs, activation function, size of hidden layers, number of hidden layers (DNN only; in RINNs, it was set to eight), and batch size. We used a combined random and grid search approach [34,35] to find the best sets of hyperparameters with the main objective of finding the optimal balance between sparsity and cross-entropy error, as explained in the next section and in [9].

### 4.4. Ranking Models Based on Balance between Sparsity and Prediction Error

The model selection in this study was more complex than standard DNN model selection, as we needed to find a balance between the sparsity of the model (i.e., sets of weight matrices for each trained network) and prediction error. In contrast, many DNNs are trained by simply finding the model with the lowest prediction error on a hold-out dataset. As was shown in previous work by our group [9], RINN models trained on simulated data with relatively high sparsity and low cross-entropy error were able to recover much of the latent causal structure contained within the data. We followed the same procedure as in our previous work [9] for selecting the best models (i.e., the models with the highest chance of containing correct causal structures). In brief, we plotted prediction error versus sparsity and measured the Euclidean distance from the origin to a set of hyperparameters, i.e., a unique, trained neural network (blue circles in the model selection figure). This distance was measured according to
(4)$$\begin{array}{c} d_{x}=\sqrt{{\left(\sum_{i=1}^{m}\sum_{j=1}^{r_{i}}\sum_{k=1}^{c_{i}}\left|w_{j,k}^{\left(i\right)}\right|\right)}^{2}+L^{2}} \end{array}$$
where *L* is the validation set’s cross-entropy loss for neural network *x*; *m* is the number of weight matrices in neural network *x*; $r_{i}$ and $c_{i}$ are the numbers of rows and columns in matrix *i*, respectively; and *w* is a scalar weight. The sets of hyperparameters were ranked according to the shortest distance from the origin (the smallest $d_{x}$); then, we retrained the models on all data for further analysis of the learned weights. Please see [9] for a more detailed explanation.

### 4.5. AUROC and Other Metrics

Area under the receiver operating characteristics (AUROC) values were calculated for each DEG and then averaged over the three validation sets, leading to 5259 AUROC values for each classifier when plotted as a histogram of AUROC values. *k*-Nearest neighbors (*k*NN) was performed using sklearn’s KNeighborsClassifier [36]). Both Euclidean and Jaccard distance metrics were evaluated, and the best *k* values were 21 and 45, respectively. Other distance metrics were evaluated with results worse than those obtained using the Jaccard distance metric. For random control, we sampled predictions from a uniform distribution over the interval [0,1) for each of the validation sets. Then, we used these predictions to calculate the AUROC values for each validation set. As with the other AUROC calculations, we took the mean over the three validation sets.

When displaying a single value for a classifier’s AUROC, the AUROC values were calculated as described above; then, the mean over all DEGs was calculated. The same procedure was followed for calculating cross-entropy error, area under the precision–recall (AUPR) curve, and the sum of the absolute values of all weights. The same random predictions described above were also used to calculate cross-entropy error and AUPR.

### 4.6. Evaluation of Learned Relationships among SGAs

Throughout this work, the results from [30] were used as ground truth for comparison purposes. Specifically, Appendix A and Figure 2 were used as ground-truth causal relationships that we hypothesized the RINN may have been able to find. Appendix A is reproduced in Appendix A, with a minor formatting modification for convenience (Appendix A). Of the 372 genes in our SGA dataset, there were 35 genes that overlapped with the genes in Appendix A. Therefore, the causal relationships that we could find were limited to the relationships among these 35 genes (Table 6).

### 4.7. SGA Weight Signature

To better understand how an RINN learns to connect SGAs to hidden nodes, we only analyzed the weights going from SGA nodes to hidden nodes. To accomplish this, we generated an “SGA weight signature” for each SGA, which is the concatenation, into a single vector, of all weights for a single SGA going from that SGA to all hidden nodes in all hidden layers. This can be visualized as the concatenation of the blue weights in Figure 1 into one long vector for each SGA. The end result was an SGA weight-signature matrix of SGAs by hidden nodes. The weight signatures for a DNN were generated using only a single weight matrix, the weight matrix between the inputs and the first hidden layer (i.e., $W_{1}$), as these are the only weights in a DNN that are specific to individual SGAs. Cosine similarity between SGA weight signatures was measured using the sklearn function *cosine_similarity* [36]. Hierarchical clustering using cosine similarity and average linkage was performed on the SGA weight signatures using the seaborn python module function *clustermap*.

Cosine-similarity community detection figures were generated using Gephi [37] and the cosine similarity between SGA weight signatures. Edges represented the highest or three highest cosine similarity for a given SGA. After generating a graph where nodes were SGAs and edges were the highest cosine similarity values, we ran the *Modularity* function in Gephi to perform community detection. *Modularity* runs an algorithm based on [38,39]. Next, we partitioned and colored the graph based on the community.

### 4.8. Visualizing an RINN as a Causal Graph

Let $G_{i,j}=\left(V,E\right)$, where $G_{i,j}$ is a causal directed acyclic graph with vertices *V* and edges *E* for neural network *i* and set of SGAs *j*. For this work, *V* represents SGAs and hidden nodes, and *E* represents directed weighted edges with weight values corresponding to the weights of an RINN. The edges are directed from SGAs (input) to DEGs (output), as we know from biology that SGAs cause changes in expression. Please see [9] for further explanation of interpreting an RINN in a causal framework. If we simply interpreted any nonzero weight in a trained RINN as an edge in $G_{i,j}$, there would be hundreds of thousands to millions of edges in the causal graph (depending on the size of the hidden layers), as $L_{1}$ regularization encourages weight values toward zero, but weights are often not actually zero. This means that some thresholding of the weights is required [9].

Even after selecting models based on the best balance between sparsity and error (the smallest $d_{x}$), our weight matrices were still very dense in terms of what could be readily visually interpreted (even after rounding weights to zero decimals). Therefore, a threshold weight value was needed to limit weight visualizations to only the largest (and, we suspect, most causally important) weights. To accomplish this, we first limited the weights to be visualized to only those that were descendants (i.e., downstream) of any of the 35 SGAs from Appendix A. Next, for each of the top ten RINN models (the ten shortest $d_{x}$), we found the absolute-value weight threshold, which led to 300 edges in total, including all SGA-to-hidden and hidden-to-hidden edges. This threshold varied slightly from model to model, ranging from 0.55 to 0.71. This threshold was chosen as it seemed to give a biologically reasonable density of edges, allowed us to recover some of the relationships in Appendix A, and was low enough to allow at least some of the causal paths to proceed from the input all the way to the output.

After finding a threshold for each model, any weight whose absolute value was greater than the threshold was added as an edge to causal graph $G_{i,j}$ for that model. The causal graphs were then plotted as modified bipartite graphs with SGAs on one side and all hidden nodes on the other. Hidden-to-hidden edges were included as arcing edges on the outside of the bipartite graph. We labeled the hidden nodes using a recursive algorithm that found all ancestor or upstream SGAs (i.e., on a path to that hidden node) for a given hidden node and graph $G_{i,j}$.

### 4.9. Finding Hidden Nodes Encoding Similar Information with Respect to SGAs

To determine if hidden nodes with similar connectivity patterns with respect to the input SGAs were shared across the best models, we needed a method to map the hidden nodes to some meaningful label. To accomplish this, we used the same recursive algorithm (described at the end of the previous section) to map each hidden node in causal graph $G_{i,j}$ to the set of SGAs that were ancestors of that hidden node. For this mapping, only the SGAs in set *j* could be used to label a hidden node, as these were the only SGAs in $G_{i,j}$. We performed this mapping using graphs generated with set *j* to only the SGAs in the individual pathways from Appendix A (e.g., j={AKT1,PIK3CA,PIK3R1,PTEN,STK11} for the PI3K pathway). For each model in the best ten models and each pathway in Appendix A, we mapped hidden nodes to a set of SGAs. Next, we compared the labeled hidden nodes across models and determined the number of models that shared identically labeled hidden nodes.

We compared the number of models that shared specific hidden nodes with random controls. Random controls were performed in the same way as described above for the experimental results, except that *j* was not set to the SGAs in one of the pathways in Appendix A—rather *j* was randomly selected from the set of SGAs including all 372 SGAs minus the 35 SGAs in Appendix A. Then, ten $G_{i,j}$, one for each of the top ten RINN models, were generated using the random SGAs. The number of models with shared hidden nodes was recorded. This procedure was repeated 30 times for each possible number of SGAs in *j*. For example, the PI3K pathway (in Appendix A) has five SGAs in it that were also in our SGA dataset. To perform random control for PI3K, we performed 30 replicates of randomly selecting five SGAs (from the set of SGAs described above) and then recorded the mean number of models sharing an *n*-SGA labeled hidden node, where *n* is the number of SGAs that a hidden node was mapped to using our recursive algorithm for finding ancestors.

## 5. Conclusions

In summary, RINN could capture the statistical relationships between genomic alterations and transcriptomic events in tumor cells. It also exhibits the ability to identify functional similarities among genes that impact the same pathways. Furthermore, RINN has the capability to uncover known relationships within cellular signaling systems and can infer hierarchical relationships of cellular signals affected by distinct SGAs.

## Figures and Tables

**Figure 1 cancers-15-03857-f001:**
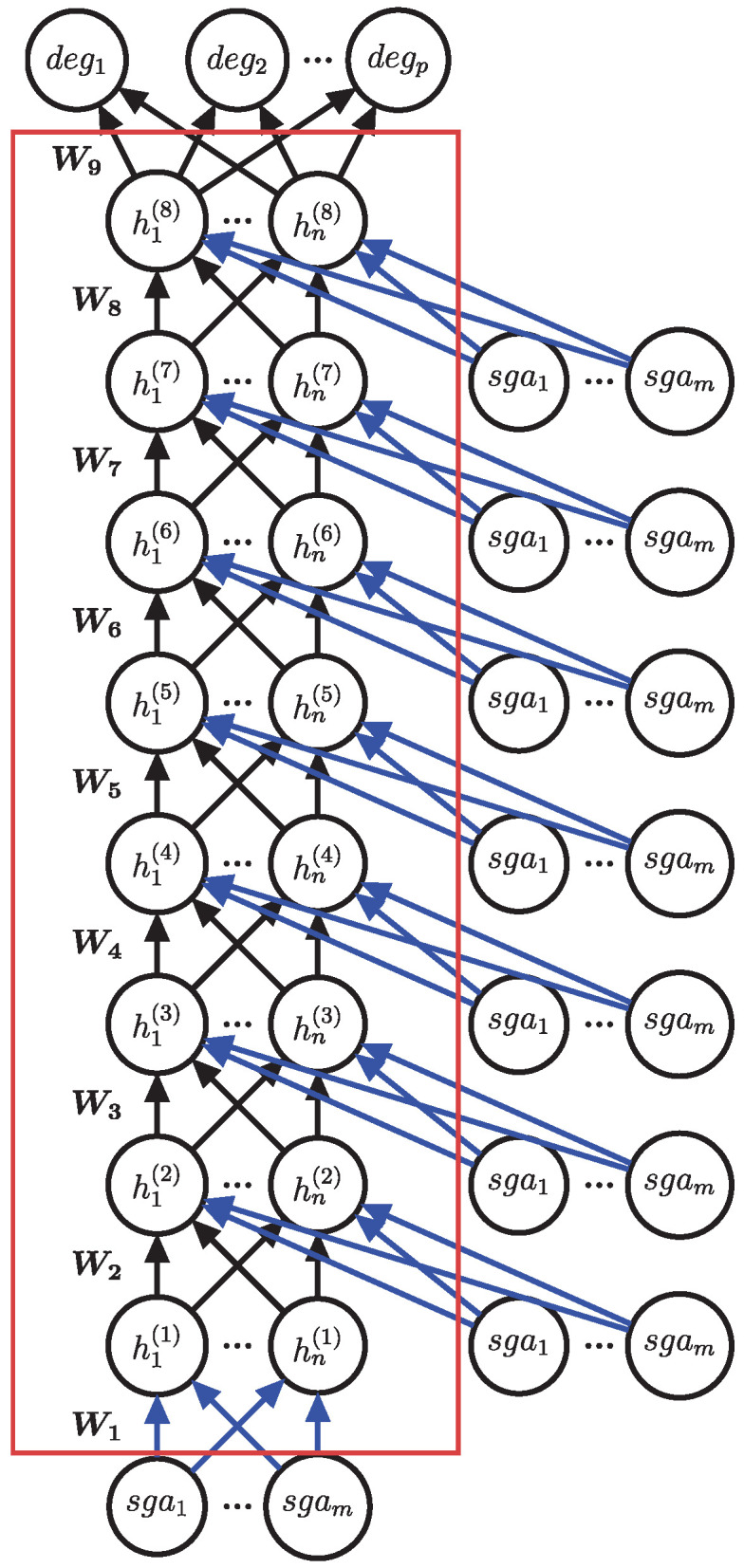
Redundant-input neural network (RINN) for TCGA data. An RINN with eight hidden layers ($h^{\left(1\right)}$, ..., $h^{\left(8\right)}$, each with *n* nodes), *m* inputs ($sga_{1}$, ..., $sga_{m}$ at the bottom), *p* outputs ($deg_{1}$, ..., $deg_{p}$), and seven sets of redundant inputs ($sga_{1}$, ..., $sga_{m}$ on the right side). Each node represents a scalar value, and each edge represents a scalar weight. The weights between layers are collected in weight matrices $W_{1}$, ..., $W_{9}$. Blue edges represent the weights used to create SGA weight signatures. The red box encloses the hidden nodes of the deep neural network.

**Figure 2 cancers-15-03857-f002:**
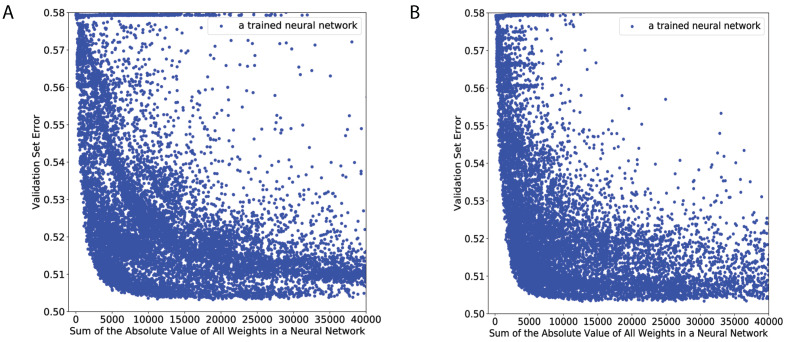
Model selection results. (**A**) RINN. (**B**) DNN.

**Figure 3 cancers-15-03857-f003:**
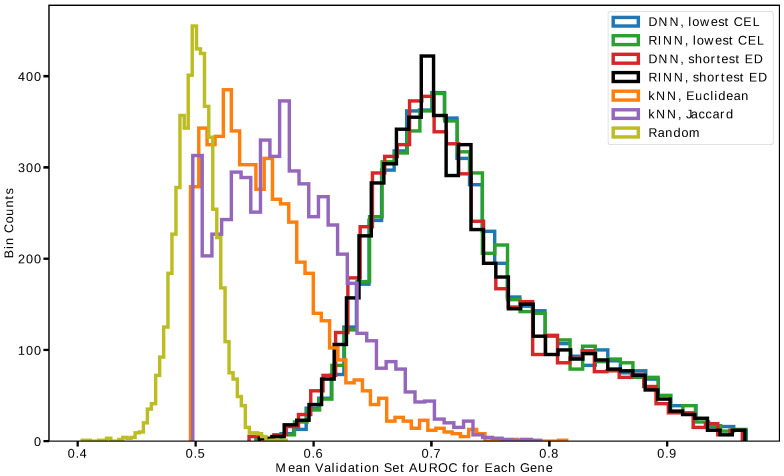
AUROC values for RINNs and DNNs. The AUROC values for predicting individual DEGs with different models are shown as histograms.

**Figure 4 cancers-15-03857-f004:**
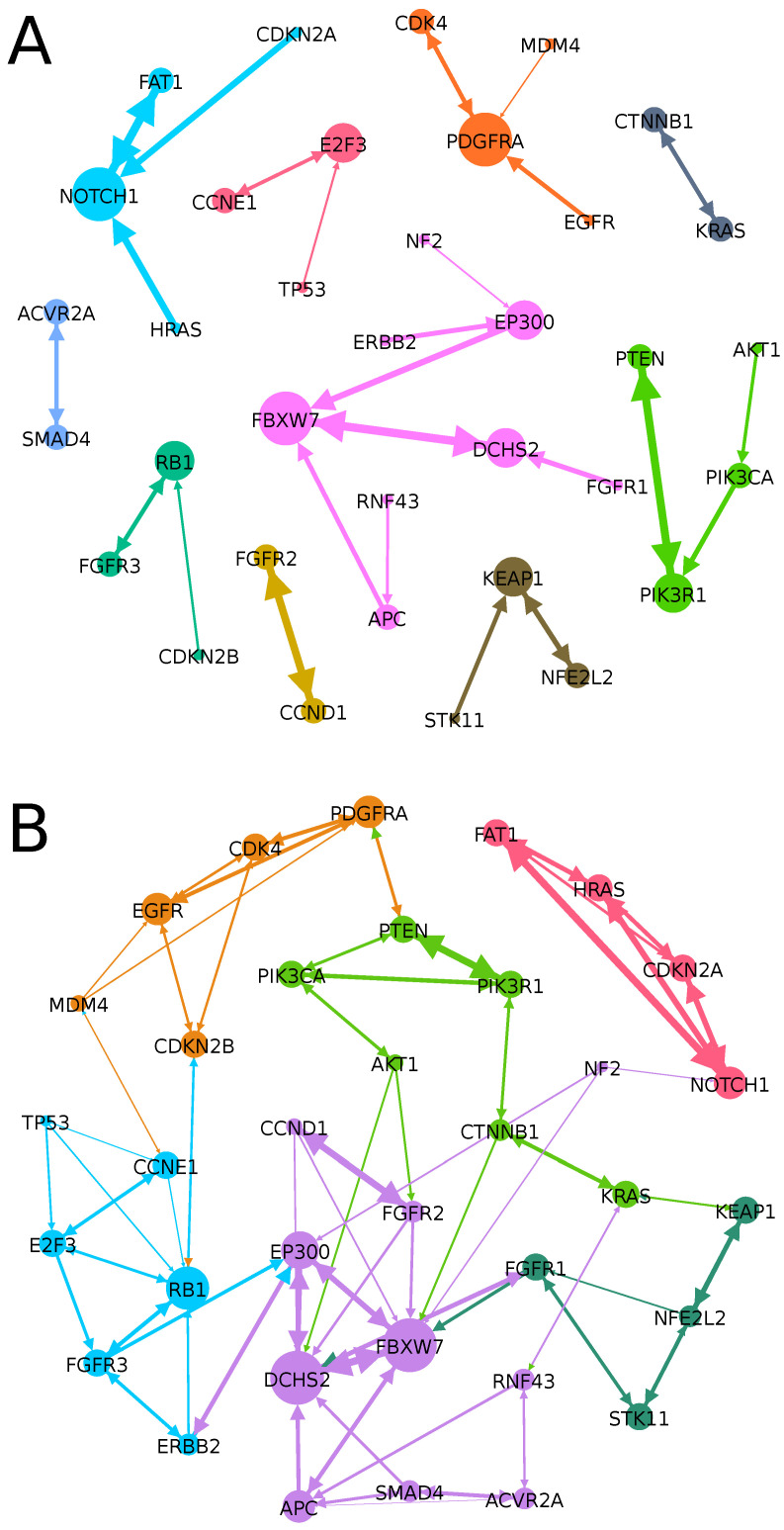
Cosine similarity of SGA weight signatures and community detection. These edges do not represent causal relationships but rather cosine-similarity (CS) relationships between SGAs, where the head of an edge indicates an SGA with high CS relative to the SGA at the tail. (**A**) Edges represent the highest cosine similarity for each SGA. (**B**) Edges represent the three highest cosine-similarity results for each SGA.

**Figure 5 cancers-15-03857-f005:**
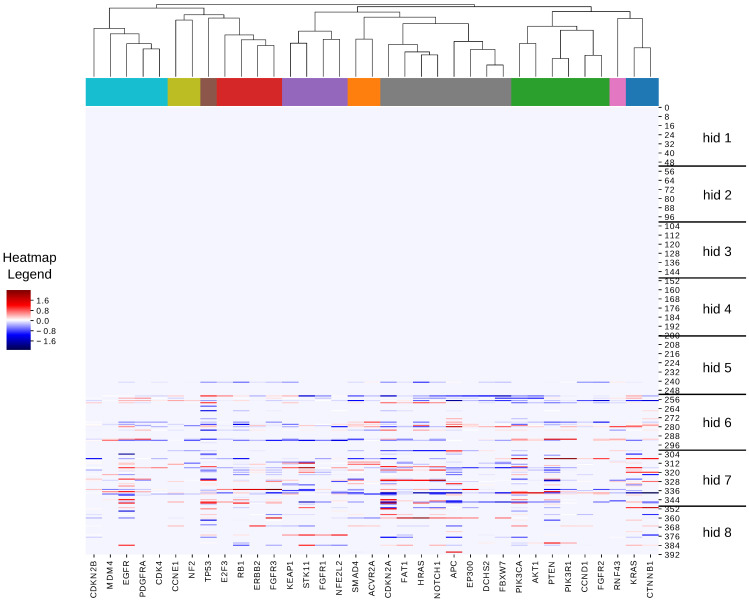
Hierarchical clustering of RINN with the shortest Euclidean distance. The heatmap shows all SGA-to-hidden-node weight values for our best trained RINN model. The vertical axis represents the hidden-node number (nodes are numbered in ascending order starting from hidden layer 1—the closest one to SGAs) and has horizontal lines representing different hidden-layer boundaries. The horizontal axis represents the 35 SGAs from Appendix A. The dendrogram and coloring at the top show the clustering of the SGAs for one particular clustering cutoff.

**Figure 6 cancers-15-03857-f006:**
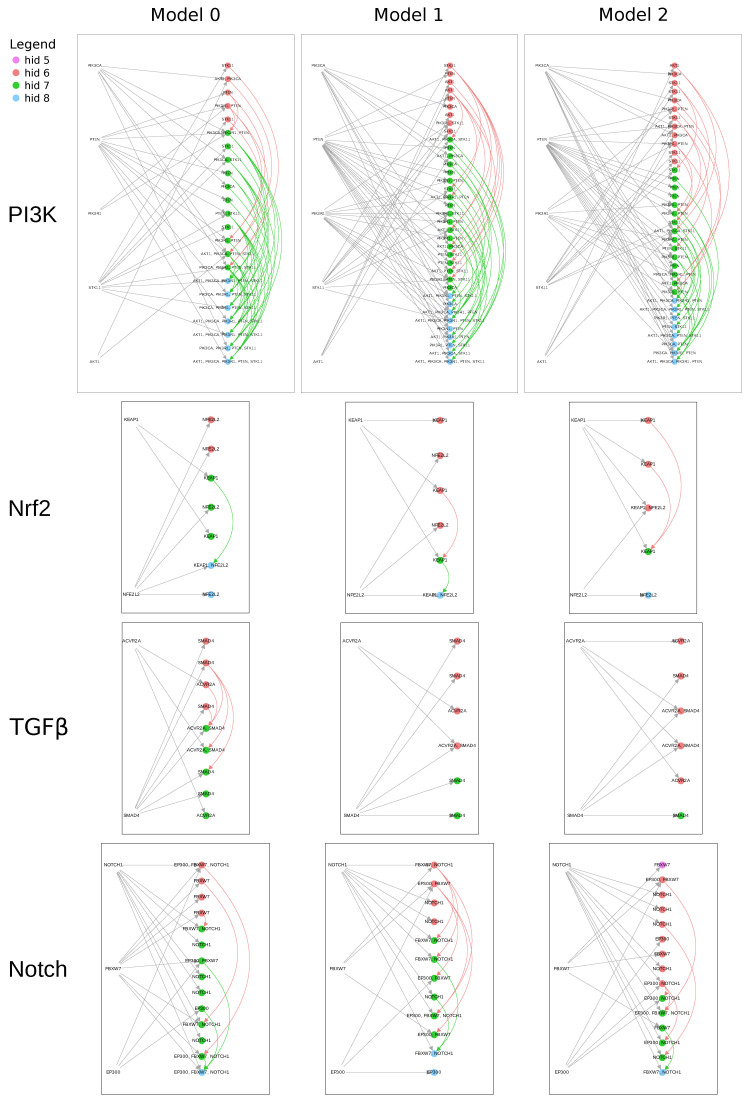
Visualizing the weights of an RINN as causal graphs. Each graph represents the causal relationships for only the SGAs on the left of the graph.

**Table 1 cancers-15-03857-t001:** Best RINN hyperparameters.

Rank	Training Epochs	Nodes in Hidden Layers	LR	RR	BS	Activation	CEL	$\sum_{1}^{8}\left|W_{i}\right|$
1	303	50	1 × 10${}^{-3}$	5 × 10${}^{-6}$	135	softplus	0.5073	4248
2	144	100	1 × 10${}^{-3}$	6 × 10${}^{-6}$	45	softplus	0.5081	3874
3	738	319	1.64 × 10${}^{-4}$	6.37 × 10${}^{-6}$	30	softplus	0.5089	3373
4	428	326	2.37 × 10${}^{-4}$	5.59 × 10${}^{-6}$	30	softplus	0.5079	4042
5	136	50	1 × 10${}^{-3}$	6 × 10${}^{-6}$	30	softplus	0.5082	4021
6	311	50	1 × 10${}^{-3}$	6 × 10${}^{-6}$	135	softplus	0.5090	3596
7	713	148	1.1 × 10${}^{-4}$	5.17 × 10${}^{-6}$	30	softplus	0.5083	4038
8	326	100	1 × 10${}^{-3}$	6 × 10${}^{-6}$	170	softplus	0.5088	3726
9	149	101	5.23 × 10${}^{-4}$	4.49 × 10${}^{-6}$	30	softplus	0.5057	5030
10	345	180	3.02 × 10${}^{-4}$	5.66 × 10${}^{-6}$	30	softplus	0.5085	3925

Hyperparameters for the ten RINN models with the lowest Euclidean distance from the origin. Each row in the table represents the set of hyperparameters that was used to fully train an RINN model. Also included are the cross-entropy errors and sum of the absolute values of the weights. ($W_{i}$ is a weight matrix between hidden layers. LR: learning rate; RR: regularization rate; BS: batch size; CEL: cross-entropy loss.)

**Table 2 cancers-15-03857-t002:** Cross-validation model selection results.

	Description	CEL	AUROC	AUPR	$\sum_{1}^{8}\left|W_{i}\right|$
**RINN 1**	Lowest CEL	**0.5032** ± 0.0048	**0.7316** ± 0.0051	**0.5678** ± 0.0039	18,899 ± 250
**RINN 2**	Second-lowest CEL	**0.5032** ± 0.0047	**0.7317** ± 0.0048	**0.5677** ± 0.0036	16,971 ± 53
**DNN 1**	Lowest CEL	**0.5033** ± 0.0052	**0.7318** ± 0.0058	**0.5676** ± 0.0041	24,503 ± 124
**DNN 2**	Second-lowest CEL	**0.5033** ± 0.0049	**0.7307** ± 0.0056	0.5663 ± 0.0032	12,739 ± 75
**RINN ED 1**	Shortest ED	0.5073 ± 0.0054	0.7258 ± 0.0054	0.5607 ± 0.0040	**4248** ± 18
**RINN ED 2**	Second-shortest ED	0.5081 ± 0.0058	0.7231 ± 0.0057	0.5569 ± 0.0038	**3874** ± 48
**DNN ED 1**	Shortest ED	0.5075 ± 0.0055	0.7238 ± 0.0057	0.5570 ± 0.0029	**3491** ± 18
**DNN ED 2**	Second-shortest ED	0.5073 ± 0.0054	0.7240 ± 0.0054	0.5576 ± 0.0031	**3704** ± 45
* **k** * **-NN**	k=21, Euclidean	9.6289 ± 0.1889	0.5599 ± 0.0034	0.5396 ± 0.0022	NA
* **k** * **-NN**	k=45, Jaccard	9.0740 ± 0.1579	0.5799 ± 0.0028	**0.5719** ± 0.0027	NA
**Random**	Preds∼U(0,1)	0.9994 ± 0.0003	0.5003 ± 0.0003	0.3257 ± 0.0052	NA

Comparison of the two RINN and DNN models with the lowest mean cross-entropy validation set loss and shortest Euclidean distance from the origin after evaluating ∼23,000 different sets of hyperparameters. Values represent the means across three validation sets. The four best models for each metric are in bold. ($W_{i}$ is a weight matrix between hidden layers. CEL: cross-entropy loss; ED: Euclidean distance.)

**Table 3 cancers-15-03857-t003:** RINN top three SGA weight-signature cosine-similarity (CS) results.

*x*	Highest CS (Mean ± SD)	Second-Highest CS	Third-Highest CS
PTEN	**PIK3R1** (0.8 ± 0.02)	PDGFRA (0.4 ± 0.01)	**PIK3CA** (0.4 ± 0.05)
PIK3CA	**PIK3R1** (0.6 ± 0.04)	**AKT1** (0.4 ± 0.06)	**PTEN** (0.5 ± 0.05)
PIK3R1	**PTEN** (0.8 ± 0.02)	**PIK3CA** (0.6 ± 0.04)	CTNNB1 (0.4 ± 0.05)
AKT1	**PIK3CA** (0.4 ± 0.06)	FGFR2 (0.4 ± 0.03)	DCHS2 (0.3 ± 0.09)
STK11	KEAP1 (0.5 ± 0.05)	FGFR1 (0.5 ± 0.05)	NFE2L2 (0.4 ± 0.10)
KEAP1	**NFE2L2** (0.6 ± 0.09)	STK11 (0.5 ± 0.05)	KRAS (0.4 ± 0.08)
NFE2L2	**KEAP1** (0.6 ± 0.09)	STK11 (0.4 ± 0.10)	FGFR1 (0.3 ± 0.11)
ACVR2A	**SMAD4** (0.5 ± 0.04)	RNF43 (0.4 ± 0.20)	APC (0.2 ± 0.10)
SMAD4	**ACVR2A** (0.5 ± 0.04)	APC (0.4 ± 0.06)	DCHS2 (0.4 ± 0.02)
CDKN2A	NOTCH1 (0.7 ± 0.09)	HRAS (0.5 ± 0.07)	FAT1 (0.4 ± 0.10)
CDKN2B	**RB1** (0.4 ± 0.14)	EGFR (0.4 ± 0.18)	**CDK4** (0.4 ± 0.18)
CCNE1	**E2F3** (0.5 ± 0.23)	**RB1** (0.3 ± 0.20)	MDM4 (0.3 ± 0.20)
CCND1	FGFR2 (0.7 ± 0.05)	DCHS2 (0.3 ± 0.11)	FBXW7 (0.3 ± 0.14)
CDK4	PDGFRA (0.5 ± 0.10)	EGFR (0.4 ± 0.11)	**CDKN2B** (0.4 ± 0.18)
RB1	FGFR3 (0.5 ± 0.07)	**E2F3** (0.4 ± 0.04)	**CDKN2B** (0.4 ± 0.14)
E2F3	**CCNE1** (0.5 ± 0.23)	FGFR3 (0.4 ± 0.10)	**RB1** (0.4 ± 0.04)
EGFR	**PDGFRA** (0.5 ± 0.10)	CDK4 (0.4 ± 0.11)	CDKN2B (0.4 ± 0.18)
FGFR1	DCHS2 (0.6 ± 0.11)	STK11 (0.5 ± 0.05)	FBXW7 (0.5 ± 0.07)
ERBB2	EP300 (0.6 ± 0.08)	**FGFR3** (0.5 ± 0.21)	RB1 (0.4 ± 0.08)
FGFR2	CCND1 (0.7 ± 0.05)	FBXW7 (0.4 ± 0.09)	DCHS2 (0.4 ± 0.11)
FGFR3	RB1 (0.5 ± 0.07)	EP300 (0.5 ± 0.02)	**ERBB2** (0.5 ± 0.21)
PDGFRA	CDK4 (0.5 ± 0.10)	**EGFR** (0.5 ± 0.10)	PTEN (0.4 ± 0.10)
KRAS	CTNNB1 (0.5 ± 0.06)	KEAP1 (0.4 ± 0.08)	RNF43 (0.3 ± 0.15)
HRAS	NOTCH1 (0.7 ± 0.05)	FAT1 (0.6 ± 0.05)	CDKN2A (0.5 ± 0.07)
MDM4	PDGFRA (0.3 ± 0.14)	EGFR (0.3 ± 0.16)	CCNE1 (0.3 ± 0.10)
TP53	E2F3 (0.4 ± 0.03)	RB1 (0.3 ± 0.08)	CCNE1 (0.3 ± 0.09)
RNF43	**APC** (0.4 ± 0.03)	ACVR2A (0.4 ± 0.20)	KRAS (0.3 ± 0.15)
APC	FBXW7 (0.5 ± 0.06)	DCHS2 (0.5 ± 0.04)	SMAD4 (0.4 ± 0.06)
CTNNB1	KRAS (0.5 ± 0.06)	PIK3R1 (0.4 ± 0.08)	FBXW7 (0.4 ± 0.07)
DCHS2	FBXW7 (0.8 ± 0.04)	EP300 (0.6 ± 0.04)	FGFR1 (0.6 ± 0.11)
FAT1	NOTCH1 (0.8 ± 0.07)	HRAS (0.6 ± 0.05)	CDKN2A (0.4 ± 0.10)
NF2	EP300 (0.3 ± 0.16)	FBXW7 (0.3 ± 0.10)	NOTCH1 (0.3 ± 0.03)
FBXW7	DCHS2 (0.8 ± 0.04)	**EP300** (0.6 ± 0.08)	APC (0.5 ± 0.06)
NOTCH1	FAT1 (0.8 ± 0.07)	HRAS (0.7 ± 0.05)	CDKN2A (0.7 ± 0.09)
EP300	**FBXW7** (0.6 ± 0.08)	DCHS2 (0.6 ± 0.04)	ERBB2 (0.6 ± 0.08)

SGAs from Appendix A with the highest cosine similarity relative to SGA *x* (mean CS across three best models ± st. dev.). SGAs in bold are in the same pathway (according to [30]) as SGA *x*.

**Table 4 cancers-15-03857-t004:** DNN top three SGA weight-signature cosine-similarity (CS) results.

*x*	Highest CS (Mean ± SD)	Second-Highest CS	Third-Highest CS
PTEN	**PIK3R1** (0.7 ± 0.06)	FGFR2 (0.5 ± 0.05)	PDGFRA (0.4 ± 0.03)
PIK3CA	**AKT1** (0.7 ± 0.06)	FGFR2 (0.5 ± 0.02)	CCND1 (0.5 ± 0.10)
PIK3R1	FGFR2 (0.7 ± 0.07)	**PTEN** (0.7 ± 0.06)	CTNNB1 (0.6 ± 0.02)
AKT1	**PIK3CA** (0.7 ± 0.06)	FGFR2 (0.5 ± 0.02)	FGFR1 (0.4 ± 0.14)
STK11	KEAP1 (0.6 ± 0.02)	FGFR1 (0.4 ± 0.12)	NFE2L2 (0.4 ± 0.01)
KEAP1	**NFE2L2** (0.7 ± 0.03)	STK11 (0.6 ± 0.02)	KRAS (0.4 ± 0.03)
NFE2L2	**KEAP1** (0.7 ± 0.03)	STK11 (0.4 ± 0.01)	NOTCH1 (0.4 ± 0.04)
ACVR2A	RNF43 (0.6 ± 0.03)	**SMAD4** (0.5 ± 0.03)	CCNE1 (0.4 ± 0.15)
SMAD4	APC (0.5 ± 0.04)	FBXW7 (0.5 ± 0.02)	**ACVR2A** (0.5 ± 0.03)
CDKN2A	NOTCH1 (0.8 ± 0.06)	**CDK4** (0.7 ± 0.02)	HRAS (0.6 ± 0.02)
CDKN2B	**CDK4** (0.6 ± 0.04)	**RB1** (0.5 ± 0.07)	EGFR (0.4 ± 0.03)
CCNE1	**E2F3** (0.6 ± 0.08)	TP53 (0.4 ± 0.07)	ACVR2A (0.4 ± 0.15)
CCND1	FGFR2 (0.6 ± 0.06)	PIK3CA (0.5 ± 0.10)	CTNNB1 (0.5 ± 0.12)
CDK4	**CDKN2A** (0.7 ± 0.02)	**CDKN2B** (0.6 ± 0.04)	EGFR (0.5 ± 0.03)
RB1	FGFR3 (0.5 ± 0.05)	TP53 (0.5 ± 0.02)	**CDKN2B** (0.5 ± 0.07)
E2F3	**CCNE1** (0.6 ± 0.08)	FGFR3 (0.5 ± 0.06)	**RB1** (0.4 ± 0.06)
EGFR	**PDGFRA** (0.6 ± 0.05)	CDK4 (0.5 ± 0.03)	MDM4 (0.5 ± 0.06)
FGFR1	FBXW7 (0.5 ± 0.10)	DCHS2 (0.5 ± 0.18)	STK11 (0.4 ± 0.12)
ERBB2	EP300 (0.6 ± 0.03)	**FGFR3** (0.5 ± 0.03)	DCHS2 (0.5 ± 0.10)
FGFR2	PIK3R1 (0.7 ± 0.07)	CCND1 (0.6 ± 0.06)	CTNNB1 (0.6 ± 0.04)
FGFR3	**ERBB2** (0.5 ± 0.03)	**HRAS** (0.5 ± 0.07)	EP300 (0.5 ± 0.08)
PDGFRA	**EGFR** (0.6 ± 0.05)	MDM4 (0.5 ± 0.14)	PTEN (0.4 ± 0.03)
KRAS	CTNNB1 (0.6 ± 0.07)	APC (0.5 ± 0.04)	SMAD4 (0.5 ± 0.02)
HRAS	NOTCH1 (0.7 ± 0.03)	CDKN2A (0.6 ± 0.02)	EP300 (0.6 ± 0.10)
MDM4	PDGFRA (0.5 ± 0.14)	EGFR (0.5 ± 0.06)	**CDKN2B** (0.3 ± 0.17)
TP53	RB1 (0.5 ± 0.02)	CCNE1 (0.4 ± 0.07)	KRAS (0.4 ± 0.06)
RNF43	ACVR2A (0.6 ± 0.03)	**APC** (0.4 ± 0.02)	KRAS (0.3 ± 0.03)
APC	FBXW7 (0.6 ± 0.02)	SMAD4 (0.5 ± 0.04)	KRAS (0.5 ± 0.04)
CTNNB1	PIK3R1 (0.6 ± 0.02)	FGFR2 (0.6 ± 0.04)	KRAS (0.6 ± 0.07)
DCHS2	FBXW7 (0.7 ± 0.08)	EP300 (0.6 ± 0.06)	FGFR1 (0.5 ± 0.18)
FAT1	NOTCH1 (0.8 ± 0.04)	CDKN2A (0.6 ± 0.09)	HRAS (0.5 ± 0.05)
NF2	EP300 (0.4 ± 0.05)	**FAT1** (0.4 ± 0.05)	NOTCH1 (0.3 ± 0.04)
FBXW7	DCHS2 (0.7 ± 0.08)	**EP300** (0.7 ± 0.09)	APC (0.6 ± 0.02)
NOTCH1	CDKN2A (0.8 ± 0.06)	FAT1 (0.8 ± 0.04)	HRAS (0.7 ± 0.03)
EP300	**FBXW7** (0.7 ± 0.09)	ERBB2 (0.6 ± 0.03)	DCHS2 (0.6 ± 0.05)

SGAs from Appendix A with the highest cosine similarity relative to SGA *x* (mean CS across three best models ± st. dev.). SGAs in bold are in the same pathway (according to [30]) as SGA *x*.

**Table 5 cancers-15-03857-t005:** Cancer pathway hidden nodes shared across top ten models.

Pathway	Mapping of Hidden Node *x*	Models (Out of 10; Numbered 0 to 9) That Shared Hidden Node *x*	Expected Number of Models that Shared an *n*-SGA Hidden Node Given *m* Random SGAs
**PI3K**	AKT1, PIK3CA, PIK3R1, PTEN, STK11	0, 1, 2, 3, 4, 5, 6, 7, 8, 9	0.0 ± 0.0 (*n* = 5, *m* = 5)
	AKT1, PIK3CA, PTEN, STK11	0, 2, 5, 6, 7, 9	0.4 ± 1.4 (*n* = 4, *m* = 5)
	PIK3CA, PIK3R1, PTEN, STK11	0, 4, 5, 7, 8	0.4 ± 1.4 (*n* = 4, *m* = 5)
	AKT1, PIK3CA, PIK3R1, PTEN	1, 3, 6	0.4 ± 1.4 (*n* = 4, *m* = 5)
	AKT1, PIK3R1, PTEN, STK11	2, 3	0.4 ± 1.4 (*n* = 4, *m* = 5)
	AKT1, PIK3CA, STK11	1, 2, 3, 5, 6, 8	1.0 ± 1.4 (*n* = 3, *m* = 5)
	PIK3R1, PTEN, STK11	2, 3, 6, 7, 8, 9	1.0 ± 1.4 (*n* = 3, *m* = 5)
	PIK3CA, PIK3R1, PTEN	0, 2, 6, 7, 9	1.0 ± 1.4 (*n* = 3, *m* = 5)
	AKT1, PIK3R1, PTEN	3, 9	1.0 ± 1.4 (*n* = 3, *m* = 5)
	AKT1, PTEN, STK11	2, 3	1.0 ± 1.4 (*n* = 3, *m* = 5)
	PTEN, STK11	0, 2, 3, 5, 6, 7, 8	3.4 ± 2.0 (*n* = 2, *m* = 5)
	PIK3R1, PTEN	0, 2, 3, 6, 8, 9	3.4 ± 2.0 (*n* = 2, *m* = 5)
	AKT1, PIK3CA	0, 3, 4, 6, 8	3.4 ± 2.0 (*n* = 2, *m* = 5)
	PIK3CA, STK11	0, 1, 2, 4, 8	3.4 ± 2.0 (*n* = 2, *m* = 5)
	AKT1, STK11	1, 2, 3, 4, 5	3.4 ± 2.0 (*n* = 2, *m* = 5)
	PIK3CA, PTEN	5, 6, 8, 9	3.4 ± 2.0 (*n* = 2, *m* = 5)
	AKT1, PIK3R1	2, 5	3.4 ± 2.0 (*n* = 2, *m* = 5)
	PIK3R1, STK11	3, 9	3.4 ± 2.0 (*n* = 2, *m* = 5)
	AKT1, PTEN	5, 9	3.4 ± 2.0 (*n* = 2, *m* = 5)
**Nrf2**	KEAP1, NFE2L2	0, 1, 2, 3, 4, 5, 6, 7, 8, 9	0.7 ± 1.2 (*n* = 2, *m* = 2)
**TGFβ**	ACVR2A, SMAD4	0, 1, 2, 3, 4, 5, 6, 7, 8, 9	0.7 ± 1.2 (*n* = 2, *m* = 2)
**Notch**	EP300, FBXW7, NOTCH1	0, 1, 2, 3, 4, 5, 6, 7, 9	0.2 ± 0.6 (*n* = 3, *m* = 3)
	FBXW7, NOTCH1	0, 2, 3, 6, 7, 8, 9	2.1 ± 2.2 (*n* = 2, *m* = 3)
	EP300, FBXW7	0, 2, 3, 4, 6	2.1 ± 2.2 (*n* = 2, *m* = 3)

*Mapping of Hidden Node x* represents SGAs that directly acted on hidden node *x* or were ancestors of hidden node *x*. Models are numbered 0 to 9. (*n*: number of SGAs mapping to a hidden node; *m*: number of SGAs in a pathway.)

**Table 6 cancers-15-03857-t006:** Genes in SGA dataset that overlap with [30].

Pathway from Appendix A	Genes in SGA Dataset and Appendix A
RTK/RAS	EGFR, FGFR1, ERBB2, FGFR2, FGFR3, PDGFRA, KRAS, HRAS
Nrf2	KEAP1, NFE2L2
TGFβ	ACVR2A, SMAD4
PI3K	PTEN, PIK3CA, PIK3R1, AKT1, STK11
p53	MDM2, CDKN2A, TP53
Cell cycle	CDKN2A, CDKN2B, CCNE1, CCND1, CDK4, RB1, E2F3
Notch	NOTCH1, FBXW7, EP300
Hippo	DCHS2, FAT1, NF2
Wnt	RNF43, APC, CTNNB1

## Data Availability

The data was gathered and analyzed following the same procedures as those employed by Cai et al. [32].

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
