# Peer review of "Revealing the Impact of Genomic Alterations on Cancer Cell Signaling with an Interpretable Deep Learning Model"

_cancers, 2023, doi:10.3390/cancers15153857_

Round 1

Reviewer 1 Report

The study utilized a deep neural network (DNN)to predict expression data from genomic alteration data (SGAs). The proposed model can recover causal relationships with the weights of a DNN. The novel approach provides a great addition to cancer mechanistic studies, and is valuable to the research community.

The manuscript was well written and easily read with a good logical flow. I only have a few minor comments

1. To define DEGs, was the normalization between tumor and normal samples performed? If so, please add the description in method.

2. Among the types of SGAs (non-synonymous mutation, small insert/deletion, or somatic copy number alteration), does each type contribute equally to prediction? 

3. When assessing model performance, what other quality matrix did the study use in addition to "sparsity and validation set error"? I didn't see others.

Author Response

  1. To define DEGs, was the normalization between tumor and normal samples performed? If so, please add the description to the method.

Answer: For the DEG, we used the binary form. In the manuscript (Experiment procedures/Data), we wrote the procedures to obtain DEG values “A binary differentially expressed gene (DEG) dataset was created by comparing the expression value of a gene in a tumor against a distribution of the expression values of the gene across normal samples from the same tissue of origin. A gene is deemed a DEG in a tumor if its value is outside the 2.5% percentile on either side of the normal sample distribution, and then that gene’s value is set to 1. Otherwise, the gene’s value was set to 0.”

  1. Among the types of SGAs (non-synonymous mutation, small insert/deletion, or somatic copy number alteration), does each type contribute equally to prediction? 

Answer: Yes, in our experiment, we consider they contribute equally to the SGA and then predict the DEG.

  1. When assessing model performance, what other quality matrix did the study use in addition to "sparsity and validation set error"? I didn't see others.

Answer: When we examine the model performance, we used cross-entropy loss, AUROC, and AUPR to evaluate, please see Table 2.  When compared with other models, like KNN with Euclidean or Jaccard matrix, we only show the AUROC for model performance comparison, please see Figure 3.

Reviewer 2 Report

This is an interesting manuscript describing an interpretable deep learning model linking somatic genomic alterations to gene expression patterns in human cancers.

The methodology looks solid and the results make sense. I would have to say that some of the findings are expected, but this indicates that the methodology is correct and is finding known and expected correlations. Having said that, there are interesting features. For example, links between FAT1 and NOTCH and between STK11 and PI3K.

The manuscript looks quite polished and I suspect that it has gone through some rounds of improvement in other journals. Therefore, I have no other comments.

Author Response

Thank you for taking the time to review our manuscript and for your positive feedback.